# BioRNN: Bio-Inspired Synergistic Integration of Neuromodulation and Wave Propagation in Recurrent Networks

## Abstract

Training recurrent networks that directly implement physical wave equations has been hindered by numerical instability and incompatibility with gradient-based optimization. We introduce BioRNN, a recurrent architecture that embeds two-dimensional wave propagation dynamics on a neural grid and achieves stable training via a mixed finite-difference scheme with learnable damping. Inspired by neuromodulation in biological systems, BioRNN incorporates a lightweight frequency-modulation stage that transforms inputs into oscillatory patterns, enabling the recurrent layer to exploit resonance and frequency selectivity. This combination allows BioRNN to model spatiotemporal dependencies through constructive interference while retaining theoretical guarantees of stability during backpropagation. On sequential visual (sMNIST, noisy CIFAR-10) and auditory (ESC-50) benchmarks, BioRNN achieves competitive performance across domains, with pronounced gains on frequency-rich auditory tasks and comparable accuracy on vision. This work demonstrates that integrating biologically inspired neuromodulation with physically grounded wave dynamics yields recurrent models that are both biologically grounded and reliably trainable within modern deep learning.

## 1 Introduction

During cognition, the brain uses a synergistic two-level system (Cools & Arnsten, 2022): neuro-modulators (chemical signals) adjust cortical responsiveness (Yang et al., 2013; Starkweather et al., 2018), while cortical sheets produce traveling waves with frequency-selective resonance (Haas & White, 2002; Liu et al., 2015). This interaction supports tasks like visual and auditory processing (Goldman-Rakic, 1988; Da Costa et al., 2013).

Neuromodulators such as acetylcholine and dopamine regulate cortical excitability by shaping responses to stimuli. Instead of encoding sensory content, they modulate the processing state of cortical networks, adjusting sensitivity and selectivity based on the task. This is conceptually similar to how structured signals guide neural architectures by modifying responsiveness without changing content (Cools & Arnsten, 2022; Yang et al., 2013; Starkweather et al., 2018).

Cortical sheets also exhibit structured wave propagation, shaped by the spatial layout of pyramidal neurons and horizontal fibers (Watakabe et al., 2023). Intermediate neurons support traveling waves, which are oscillatory patterns moving through space and time. These waves enable frequency-selective resonance (Fu et al., 2024; Jeon et al., 2024), helping maintain localized spatiotemporal representations (McCormick et al., 2020). These biological principles suggest that effective neural computation requires the synergistic integration of modulatory preprocessing and wave-based dynamics.

From a computational perspective, however, embedding physical wave equations directly into recurrent neural networks has long been hindered by numerical instability and incompatibility with gradient-based training. While RNNs are often used to model temporal and biological mechanisms, conventional gated RNNs diverge from physical principles, and existing bio-inspired approaches such as CoRNN (Rusch & Mishra, 2021), Neural Wave Machine (NWM) (Keller & Welling, 2023), and wRNN (Keller et al., 2024) only partially capture biological dynamics. They typically omit

modulatory preprocessing and approximate wave propagation through simplified couplings, rather than grounding dynamics in a physically principled formulation. As a result, they cannot fully reproduce the synergy between neuromodulation and resonance observed in real neural circuits.

To address these gaps, we propose **BioRNN** (Fig. 1), a biologically grounded recurrent architecture that integrates neuromodulatory preprocessing with wave dynamics implemented as a stable finite-difference time-domain (FDTD2D) system. BioRNN addresses both identified limitations through two components. First, a trainable frequency-selective input modulation module converts raw inputs into oscillatory patterns (Fig. 1(b)), which drive resonant neural dynamics across modalities. For audio, modulation enriches spectral representations with learnable frequency, phase, and mixing parameters, while for visual data it injects oscillatory structure into otherwise static pixel inputs. Second, the core hidden layer employs neurons arranged on a spatial grid (Fig. 1(c)) with dynamics governed by discretized two-dimensional wave equations, involving two neuron types and tunable parameters for wave speed $c$ and damping coefficient $k$. Together, these components create a frequency-selective landscape where adjusting $c$ and $k$ tunes spatial regions to specific spectral components, analogous to tuning a radio. Crucially, our formulation combines biological inspiration with physical wave principles while ensuring backpropagation stability via a mixed finite-difference scheme and parametric damping.

Our experiments validate the benefits of this design across both visual and auditory sequence tasks. BioRNN achieves competitive performance relative to conventional RNNs and recent bio-inspired models, with pronounced gains on frequency-rich auditory benchmarks and comparable performance on vision. We further show that distinct propagation regimes emerge depending on modality: auditory tasks favor slower, sustained waves, while visual tasks benefit from faster propagation, reflecting oscillatory behaviors observed in different brain regions. These results highlight how biological principles, combined with physical stability guarantees, enable effective resonance-based computation in deep learning.

In summary, our contributions are as follows:

- We propose **BioRNN**, a bio-inspired recurrent architecture that unifies trainable neuromodulation with wave dynamics implemented as a stable FDTD2D system.

- We provide theoretical guarantees for the numerical stability of wave-equation dynamics under backpropagation.

- We demonstrate competitive performance across vision and auditory tasks, with pronounced gains on frequency-rich auditory benchmarks and interpretable dynamics grounded in biological principles.

## 2 METHODS

The Bio-inspired Recurrent Neural Network (BioRNN), shown in Fig. 1(a), consists of two main components: an input modulation module (Fig. 1(b)) and a two-dimensional coupled recurrent hidden layer of size $G \times G$ (Fig. 1(c)).

BioRNN operates in a recurrent fashion, where the hidden state at each time step is updated based on both the current input and the previous hidden state. The update equations are defined as Eqs. equation 1:

$$\mathbf{h}_t = f\Big(\mathbf{W}_{\text{in}}\big(\mathbf{u}_t \cdot \mathcal{M}(t)\big) + \mathcal{S}(\mathbf{h}_{t-1})\Big),$$
$$out_t = \mathbf{W}_{\text{out}}\mathbf{h}_t, \tag{1}$$

where $\mathbf{u}_t \in \mathbb{R}^d$ is the input at time $t$, $\mathcal{M}(t)$ is the time-dependent modulation function, and $\tilde{\mathbf{u}}_t = \mathbf{u}_t \cdot \mathcal{M}(t)$ is the modulated input. The matrix $\mathbf{W}_{\text{in}} \in \mathbb{R}^{G^2 \times d}$ projects the modulated input onto a 2D spatial grid. The hidden state $\mathbf{h}_t \in \mathbb{R}^{G^2}$ represents primary field ($p$) distributed over the spatial domain. The function $f(\cdot)$ is a pointwise nonlinearity (e.g., $\tanh$), and $\mathcal{S}(\cdot)$ is a structured spatial operator modeling wave-like propagation across the grid. Finally, $\mathbf{W}_{\text{out}} \in \mathbb{R}^{n_{\text{out}} \times G^2}$ is the readout matrix producing the predicted output $out_t \in \mathbb{R}^{n_{\text{out}}}$.

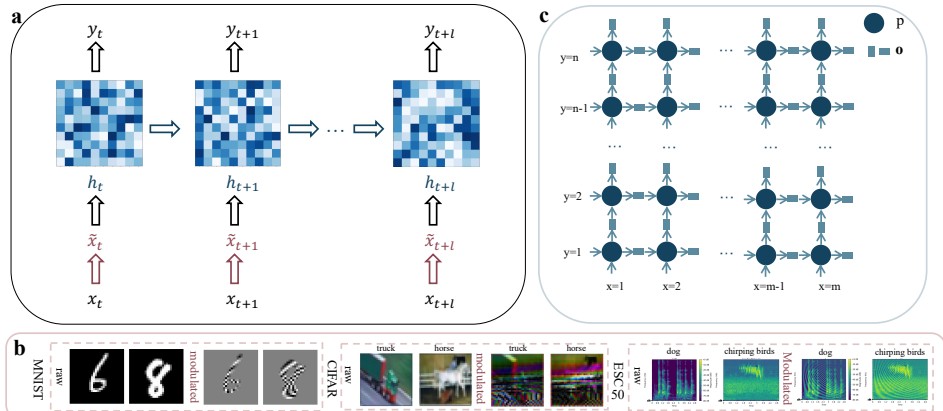

Figure 1: BioRNN processes sequential data through biologically inspired modulation and wave propagation. (a) Information flow: raw input $\mathbf{u}_t$ is converted to oscillatory patterns $\tilde{\mathbf{u}}_t$ by the modulation module, processed by the wave-based recurrent layer to produce hidden states $\mathbf{h}_t$, and classified to output $out_t$. (b) Modulation examples showing how different input types are converted to frequency-rich patterns on three datasets: MNIST (handwritten digits), noisy CIFAR (natural images), and ESC50 (environmental sounds). Visual tasks use pixel values; ESC50 uses log-Mel spectrograms from raw waveforms. Left: raw data; right: modulated input. (c) The core hidden layer: a 2D recurrent grid with two neuron types, $p$ neurons (input/output regions) and $o$ neurons (intermediate processing), enabling frequency-selective sequence processing through wave-like dynamics.

## 2.1 Input Modulator $\mathcal{M}$

To activate the frequency-selective resonant dynamics of the recurrent layer, we incorporate a sinusoidal modulation mechanism into the input preprocessing. This enriches the input with diverse frequency content, which is essential for resonance-based processing.

At initialization, each time step receives a distinct sinusoidal modulation frequency that increased with the time step index. These frequencies are treated as learnable parameters during training, allowing the model to adapt its temporal sensitivity to different tasks. Having established how raw inputs are converted to oscillatory patterns, we now describe how these patterns drive resonant dynamics in the spatially organized recurrent layer.

## 2.2 2D Coupling Structure $\mathcal{S}$ of Hidden Layer

The BioRNN's hidden layer exhibits wave-like propagation through a biologically inspired 2D structure composed of two neuron types: primary neurons ($p$), responsible for storing and representing information, and auxiliary neurons ($\mathbf{o}$), which mediate spatial transmission.

The dynamics are governed by a system of coupled first-order partial differential equations (PDEs), inspired by wave propagation in lossy media:

$$\frac{\partial p}{\partial t} + k^p p + c^2 \nabla \cdot \mathbf{o} = \mathbf{u}_t,$$
$$\frac{\partial \mathbf{o}}{\partial t} + \mathbf{k^o} \cdot \mathbf{o} + \nabla p = 0, \tag{2}$$

where $\mathbf{u}_t$ is the time-dependent input, $c$ is the propagation speed (analogous to axonal conduction velocity), and $k^p$, $\mathbf{k^o}$ are attenuation parameters (similar to membrane time constants) for $p$ and $\mathbf{o} = (o_x, o_y)$, respectively. This formulation enables both spatiotemporal memory and directional signal flow.

For efficient computation, we reparameterize $\mathbf{o}' = c\mathbf{o}$, leading to:

$$\frac{\partial p}{\partial t} + k^p p + c\nabla \cdot \mathbf{o}' = \mathbf{u}_t,$$

$$\frac{\partial \mathbf{o}'}{\partial t} + \mathbf{k^o} \cdot \mathbf{o}' + c\nabla p = 0, \tag{3}$$

where we drop the prime notation. $k^p$ provides uniform damping, while $\mathbf{k^o}$ selectively suppresses spatial variations.

## 2.3 DISCRETIZATION

### 2.3.1 NUMERICAL STABILITY OF WAVE PROPAGATION

To ensure stable simulation of wave propagation governed by PDEs, the Courant–Friedrichs–Lewy (CFL) condition needs to be satisfied. It guarantees that the numerical domain of dependence includes the physical one, which is essential for explicit time integration. In one dimension, this condition is given by CFL $= \frac{c\Delta t}{\Delta x}$ or $\frac{c\Delta t}{\Delta y} \leq 1$, where $c$ is the wave speed, $\Delta t$ the time step, $\Delta x$ and $\Delta y$ are the spatial resolution on x and y axis, respectively.

In our BioRNN, where PDE dynamics are embedded in the recurrent update of hidden state, both $c$ and $k$ are trainable. This may lead to CFL violations during training. To mitigate this, we use a mixed finite difference scheme that balances numerical dissipation and accuracy, ensuring stable dynamics even when the CFL condition is occasionally exceeded.

To implement these continuous dynamics in practice, we employ a discretization scheme that maintains numerical stability.

### 2.3.2 MIXED FINITE DIFFERENCE DISCRETIZATION

To simulate the BioRNN dynamics in Eqs. equation 3, we adopt a fully explicit finite difference scheme with first-order accuracy in space and time. Each time step consists of two phases: wave propagation and damping. In this study, we set $\Delta x = \Delta y = 1$ and, for notational simplicity, omit these variables in the subsequent derivations.

In the first phase, we compute intermediate states using forward differences:

$$o^*_{x,m,n} = o^t_{x,m,n} - \Delta t \cdot c \cdot (p^t_{m+1,n} - p^t_{m,n}),$$

$$o^*_{y,m,n} = o^t_{y,m,n} - \Delta t \cdot c \cdot (p^t_{m,n+1} - p^t_{m,n}),$$

$$\nabla \cdot \mathbf{o}^*_{m,n} = (o^*_{x,m+1,n} - o^*_{x,m,n}) + (o^*_{y,m,n+1} - o^*_{y,m,n}),$$

$$p^*_{m,n} = p^t_{m,n} - \Delta t \cdot c \cdot \nabla \cdot \mathbf{o}^*_{m,n} + \Delta t \cdot I^t_{m,n}, \tag{4}$$

where $p^t_{m,n}$ and $o^t_{x,m,n}, o^t_{y,m,n}$ denote the value of $p$ and $o$ neuron at spatial index $(m,n)$ and time step $t$, and $I^t_{m,n}$ is the input at that location and time. The superscript $*$ indicates intermediate (pre-damping) states.

In the second phase, damping is applied to the intermediate states via an implicit decay rule:

$$\mathbf{o}^{t+1}_{m,n} = \frac{\mathbf{o}^*_{m,n}}{1 + \Delta t \cdot k^{\mathbf{o}}_{m,n}}, p^{t+1}_{m,n} = \frac{p^*_{m,n}}{1 + \Delta t \cdot k^p_{m,n}}, \tag{5}$$

where $k^p_{m,n}$ and $k^{\mathbf{o}}_{m,n}$ are the damping coefficients for the $p$ and $\mathbf{o}$ at grid $(m,n)$, respectively; and $t+1$ denotes the next time step.

### 2.3.3 SPATIAL OPERATORS

We define the spatial gradient and divergence operators using finite difference approximations.

The gradient is computed using backward differences:

$$\left.\frac{\partial p}{\partial x}\right|_{m,n} \approx \frac{p_{m,n} - p_{m-1,n}}{\Delta x}, \left.\frac{\partial p}{\partial y}\right|_{m,n} \approx \frac{p_{m,n} - p_{m,n-1}}{\Delta y}, \tag{6}$$

and the divergence is computed using forward differences:

$$\left.\frac{\partial o_x}{\partial x}\right|_{m,n} \approx \frac{o_{x,m+1,n} - o_{x,m,n}}{\Delta x}, \left.\frac{\partial o_y}{\partial y}\right|_{m,n} \approx \frac{o_{y,m,n+1} - o_{y,m,n}}{\Delta y}. \tag{7}$$

## 2.4 Stability Analysis for Hidden State of BioRNN

We analyze the stability of the mixed finite difference discretization by examining how its spatiotemporal structure behaves under Fourier-mode perturbations. In particular, we demonstrate that the discretization introduces numerical dissipation that enables robust training, even when the CFL condition is exceeded. This stability arises from two complementary mechanisms: high-frequency damping induced by asymmetries in the discretization scheme and exponential decay from learnable damping parameters.

**Lemma 1** (Mixed Difference Fourier Properties). *Let $\xi_x \in [-\pi, \pi]$ denote the normalized spatial frequency in the $x$-direction, corresponding to a Fourier mode on a uniform grid with spacing $\Delta x$.*

*For a discrete derivative operator, its action in Fourier space can be represented by an effective frequency symbol $\tilde{\xi}_x$, which plays the role of the continuous derivative symbol $i\xi_x$. The specific form of $\tilde{\xi}_x$ depends on the choice of finite difference stencil.*

*The backward difference operator has Fourier symbol*

$$\tilde{\xi}_x^{backward} = i\xi_x + \frac{\xi_x^2 \Delta x}{2} + \mathcal{O}(\Delta x^2), \tag{8}$$

*while the forward difference operator has Fourier symbol*

$$\tilde{\xi}_x^{forward} = i\xi_x - \frac{\xi_x^2 \Delta x}{2} + \mathcal{O}(\Delta x^2). \tag{9}$$

*The opposite signs of the leading-order correction terms indicate that backward differences introduce numerical dissipation while forward differences introduce numerical anti-dissipation. When used together in mixed schemes, these effects balance to attenuate spurious high-frequency modes and stabilize the numerical dynamics.*

*Proof.* By applying the spatial translation theorem, the Fourier symbols of the backward and forward differences are obtained as $\frac{1-e^{-i\xi_x \Delta x}}{\Delta x}$ and $\frac{e^{i\xi_x \Delta x}-1}{\Delta x}$, respectively. Taylor expansion of $e^{\pm i\xi_x \Delta x}$ yields effective frequency symbols $i\xi_x \pm \frac{\xi_x^2 \Delta x}{2} + \mathcal{O}(\Delta x^2)$. The opposite-sign corrections correspond to numerical dissipation and anti-dissipation, whose complementary effects stabilize high-frequency modes. Full derivation is provided in Appendix B.1. □

To study how these effects propagate through the BioRNN dynamics, we linearize the discrete update equations in the frequency domain. This yields the following recurrent coupling matrix $\mathbf{A}$, which governs the evolution of the primary field $p$ and auxiliary fields $\mathbf{o}$ in spectral space (detailed derivation is provided in Appendix B.2).

$$\mathbf{A} = \begin{bmatrix} A_{11} & A_{12} & A_{13} \\ A_{21} & A_{22} & A_{23} \\ A_{31} & A_{32} & A_{33} \end{bmatrix}, \tag{10}$$

where $\alpha = \frac{1}{1+\Delta t \cdot k^p} < 1$ and $\beta = \frac{1}{1+\Delta t \cdot k^o} < 1$, and

$A_{11} = \alpha \left( 1 - (\Delta t \cdot c)^2 (\xi_x^2 + \xi_y^2) - \frac{(\Delta t \cdot c)^2}{4} (\xi_x^4 + \xi_y^4) \right),$

$A_{12} = -\alpha \Delta t \cdot c \left( i\xi_x - \frac{\xi_x^2}{2} \right), A_{13} = -\alpha \Delta t \cdot c \left( i\xi_y - \frac{\xi_y^2}{2} \right),$

$A_{21} = -\beta \Delta t \cdot c \left( i\xi_x + \frac{\xi_x^2}{2} \right), \quad A_{22} = \beta, \quad A_{23} = 0,$

$A_{31} = -\beta \Delta t \cdot c \left( i\xi_y + \frac{\xi_y^2}{2} \right), \quad A_{32} = 0, \quad A_{33} = \beta$, where $\alpha = \frac{1}{1+\Delta t \cdot k^p}$ and $\beta = \frac{1}{1+\Delta t \cdot k^o}$.

**Theorem 1** (Mixed Differencing Improves Stability Margin). *For the primary field's recurrent coupling term $A_{11}$, the mixed difference scheme enhances stability at high-frequency modes ($|\xi| \to \pi$) by introducing an additional dissipative contribution:*

$$A_{11}^{mixed} < A_{11}^{central}, \tag{11}$$

Table 1: Dataset characteristics and their respective input formats for BioRNN. Datasets span visual and auditory domains with varied sequence lengths and modalities. ESC50-10 ms and ESC50-100 ms represent the same dataset at different frame resolutions (10 ms vs. 100 ms), highlighting how input granularity affects temporal depth. Only the ESC50 variants exhibit natural frequency structure, which is essential for evaluating biologically inspired recurrent models.

| Characteristic | sMNIST | nsCIFAR-10 | ESC50-10 ms | ESC50-100 ms |
|---|---|---|---|---|
| Sequence length | 28 steps (rows) | 64 steps (patches) | 500 steps (frames) | 50 steps (frames) |
| Input modality | Visual (grayscale) | Visual (color) | Audio (spectral) | Audio (spectral) |
| Input dimensions | 28 dim/step | 48 dim/step | 128 dim/step | 128 dim/step |
| Natural frequency | ✗ | ✗ | ✓ | ✓ |

*which selectively damps grid-scale oscillations while leaving low-frequency modes essentially unaffected. As a result, mixed differencing increases the stability margin compared to pure central differencing.*

*Proof.* The mixed stencil augments the central symbol with a nonnegative high-order term that scales like $\mathcal{O}(|\xi|^4)$ as $|\xi| \to 0$, but dominates at $|\xi| \approx \pi$. Consequently, high-frequency Fourier modes are more strongly suppressed, ensuring the amplification factor is reduced relative to central differencing. Full derivation is provided in Appendix B.3. □

**Theorem 2** (Two Stabilization Mechanisms). *The BioRNN two-phase update structure achieves stability through a combination of numerical and parametric damping. The matrix $\mathbf{A}$ incorporates two stabilization effects: (1) a high-frequency dissipation term $-\frac{(\Delta t \cdot c)^2}{4}(\xi_x^4 + \xi_y^4)$ in $A_{11}$, resulting from the mixed difference operators, and (2) multiplicative attenuation via damping coefficients $k^p, k^o > 0$, encoded through scaling factors $\alpha < 1$ and $\beta < 1$. Together, these mechanisms ensure robust suppression of high-frequency instabilities in the recurrent dynamics.*

*Proof.* The stabilization mechanisms follow directly from the structure of matrix $\mathbf{A}$. The high-frequency dissipation term in $A_{11}$ arises from Lemma 1, while the multiplicative attenuation through $\alpha < 1$ and $\beta < 1$ provides additional damping across all modes. The combination ensures robust stability. Full derivation in Appendix B.2. □

## 3 EXPERIMENTS

To evaluate the performance and robustness of the proposed model, we conducted experiments on diverse benchmark tasks across multiple modalities. These tasks were selected to test the model's ability to process sequential data under different conditions, both with and without oscillatory dynamics. We compared results with several bio-inspired RNN baselines and adapted input modulation strategies to each task. Performance was analyzed under multiple parameter initializations to ensure statistical reliability. We also performed ablation studies to examine the structural roles of key components in the BioRNN. All hyperparameters and environment settings are detailed in Appendix C.

### 3.1 EVALUATED TASKS AND INPUT MODULATION STRATEGIES

We evaluated BioRNN on three tasks spanning visual and auditory modalities, each transformed into sequential input representations to align with the model's spatiotemporal dynamics. To stimulate the frequency-selective resonance behavior of BioRNN, we applied a sinusoidal input modulation of the general form: $\tilde{x}_t^i = x_t^i \cdot \mathcal{M}(i, t)$, where $\mathcal{M}(i, t)$ is a task-specific modulation function based on spatial index $i$ and time/frame index $t$, with learnable frequency parameters.

**Sequential MNIST (sMNIST).** Each $28 \times 28$ image is reshaped into a sequence of 28 row vectors $\mathbf{u}_t \in \mathbb{R}^{28}$. Modulation is applied as $\mathcal{M}(i, t) = \sin\left(2\pi f_t \cdot \frac{n_i}{D}\right)$ with $D = 28$.

**Noisy CIFAR-10 (nsCIFAR-10).** Each $32 \times 32$ RGB image is divided into 64 flattened $4 \times 4$ patches with 3 RGB channels ($\mathbf{u}_t \in \mathbb{R}^{48}$). A scaled modulation is used to maintain values in $[0,1]$: $\mathcal{M}(i,t) = 0.5 + 0.5 \cdot \sin\left(2\pi f_t \cdot \frac{n_i}{D}\right)$ with $D = 48$.

**ESC50.** Each audio clip is converted into a log Mel-spectrogram with $D = 128$ Mel-frequency bins, yielding input vectors $\mathbf{u}_t \in \mathbb{R}^D$ at each time step $t$. Modulation is applied jointly along the frequency and time axes using trainable parameters $f_i$ and $\phi_i$:

$$\mathcal{M}(i,t) \;=\; (1 - \alpha_i) \;+\; \alpha_i \cdot \frac{\sin\left(2\pi f_i \frac{t}{T} + \phi_i\right) + 1}{2},$$

where $i = 1, \ldots, D$ indexes the Mel-frequency bins, $t = 1, \ldots, T$ indexes the time frames, $f_i$ is a learnable frequency parameter controlling the oscillation rate, $\phi_i$ is a learnable phase offset, and $\alpha_i \in (0,1)$ is a learnable mixing coefficient that interpolates between the unmodulated input ($\alpha_i \to 0$) and fully modulated input ($\alpha_i \to 1$).

Table 1 summarizes input dimensionality and sequence characteristics. The modulation enhances frequency richness in the input, enabling BioRNN to engage its resonance-based dynamics across visual and auditory domains.

## 3.2 PERFORMANCE COMPARISON ACROSS TASKS

We evaluated BioRNN against both conventional and bio-inspired RNN baselines across three sequential tasks that test different aspects of frequency-selective processing. As conventional baselines, we include the LSTM (Hochreiter & Schmidhuber, 1997) and GRU (Cho et al., 2014), two widely used gated RNNs known for their effectiveness in capturing long-term dependencies.

Table 2: Classification accuracy (%) on sMNIST, nsCIFAR-10, and ESC50-10 ms. ESC-50 results are reported as 5-fold averages (full results shown in Table C.2). For Base models, no modulation variant exists. For other groups, the right block reports results *with* input modulation, with parentheses showing the change relative to no modulation.

| Model | Without mod. | | | With mod. | | |
|---|---|---|---|---|---|---|
| | sMNIST | nsCIFAR-10 | ESC50-10 ms | sMNIST | nsCIFAR-10 | ESC50-10 ms |
| LSTM | $98.8^3$ | $58.3^3$ | $32.4^3$ | 97.4 | 57.4 | 32.5 |
| GRU | $99.1^2$ | 60.8 | $42.2^1$ | 97.5 | $61.2^1$ | 42.2 |
| NWM | $98.6^4$ | $56.2^4$ | 4.0 | 94.9 (-3.7) | 54.1 (-2.1) | 5.7 (+1.7) |
| coRNN | $99.3^1$ | $59.0^2$ | 10.5 | 97.0 (-2.3) | 37.3 (-21.7) | 6.1 (-4.4) |
| wRNN | 97.6 | 55.0 | 16.8 | 95.9 (-1.7) | 16.1 (-38.9) | 17.2 (+0.4) |
| BioRNN | 11.2 | 53.4 | 32.7 | 98.1 (+86.9) | 54.2 (+0.8) | $33.6^2$ (+0.9) |

Among biologically inspired RNNs, NWM (Keller & Welling, 2023) arranges neurons in a 2D grid with oscillatory dynamics, coRNN (Rusch & Mishra, 2021) introduces second-order coupling for stable memory, and wRNN (Keller et al., 2024) treats hidden states as traveling waves. These PDE-based baselines capture oscillatory structure but show limited robustness when modulation is applied, often degrading on vision tasks and only providing modest gains on ESC50.

BioRNN, in contrast, consistently benefits from modulation across all tasks. On vision benchmarks, it remains competitive with coRNN and wRNN despite their architectural advantages. The strongest effect is observed on ESC50, where BioRNN surpasses other PDE-based models under equal neuron budgets, highlighting the effectiveness of its frequency-selective recurrent dynamics. While NWM and wRNN show small improvements from modulation in this auditory domain, BioRNN achieves a substantially higher accuracy, narrowing the gap with the best conventional baseline (GRU).

In summary, the modulator acts as a task-adaptive transformation that aligns well with BioRNN's resonant dynamics, enabling competitive vision performance and clear advantages on spectrally rich audio tasks.

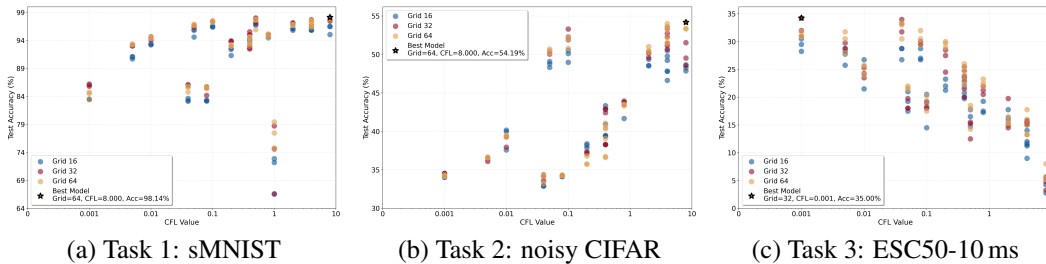

(a) Task 1: sMNIST  (b) Task 2: noisy CIFAR  (c) Task 3: ESC50-10 ms

Figure 2: Scatter plots of test accuracy (y-axis) versus CFL value (x-axis) for the three evaluated tasks. Each panel corresponds to one task: (a) sMNIST, (b) noisy CIFAR-10, and (c) ESC50-10 ms. For a fixed CFL value, the variation in accuracy is mainly attributed to differences in grid size $G$ and damping coefficients ($k^p$ and $k^o$) of the model. Based on the ANOVA analysis, $G$ is identified as a key factor influencing performance, ranking just after wave speed ($c$) and time step ($\Delta t$) in single-factor importance. Therefore, we use different colors to indicate $G = 16, 32, 64$ in the scatter plots. The five-pointed star marks the best-performing parameter combination for each task. Since the scatter distributions for ESC50-10 ms and ESC50-100 ms are highly similar, only the 10 ms version is shown.

### 3.3 Impact of Initialization on BioRNN Performance

We evaluated the impact of initialization parameters on BioRNN performance using three benchmark datasets with varying modalities and complexities: sMNIST, nsCIFAR-10, and ESC50. A comprehensive grid search was performed across time step $\Delta t \in \{0.001, 0.005, 0.01, 0.05, 0.1\}$, fixed spatial step $\Delta x = 1.0$, wave speed $c \in \{1, 40, 80\}$, damping coefficient $k \in \{0.01, 0.05, 0.1\}$ (with $k^p = k^o$), and grid size $G \in \{16, 32, 64\}$. This allowed us to analyze individual and interactive effects of parameters across tasks.

Single-factor ANOVA (Appendix D. Table 2) showed that $\Delta t$ explains most performance variance across all tasks, especially for visual datasets: 95.32% (sMNIST) and 95.67% (nsCIFAR-10). For audio tasks, the influence of $\Delta t$ dropped to 78.25% (ESC50-10 ms) and 67.94% (ESC50-100 ms), indicating greater sensitivity to complex spectral dynamics. Appendix A3. Table 2 also shows optimal $\Delta t$ values for all tasks: coarse steps ($\Delta t = 0.1$) suffice for sMNIST and nsCIFAR-10, while ESC50 requires finer steps ($\Delta t = 0.001$ for 10 ms, 0.005 for 100 ms), highlighting the importance of high temporal resolution for audio.

Multi-factor ANOVA (Appendix A3. Table 3) revealed that the interaction $c \times \Delta t$ explains the highest variance across tasks. With fixed $\Delta x$, this corresponds to the CFL condition as $\Delta x = 1$. Fig. 2 shows that visual tasks perform best at CFL $\approx 8.0$, tolerating coarse discretization. In contrast, ESC50 peaks at CFL $\approx 0.001$ and degrades rapidly beyond 0.4, reflecting the need to preserve fine temporal structure.

These CFL differences are not explained by sequence length: despite nsCIFAR-10 having more time steps than sMNIST, both share optimal CFL ranges, while ESC50-100 ms (only 50 steps) requires a stricter CFL. Appendix A3. Table 3 shows that while $\Delta t$ dominates visual tasks (90.87% for sMNIST, 95.48% for nsCIFAR-10), its contribution drops in audio (50.56% for ESC50-100 ms, 76.64% for ESC50-10 ms), with $\Delta t \times c$ interactions becoming more influential (up to 4.07% in ESC50).

Beyond CFL, secondary factors matter per task. Wave speed $c$ impacts sMNIST (15.14%), supporting long-range memory. Grid size $G$ is key for ESC50-100 ms (15.29%), enhancing spatial fidelity. Damping $k$ contributes modestly but consistently across tasks, stabilizing dynamics.

From the analysis, BioRNN initialization is best guided by task complexity rather than sequence length. Visual tasks tolerate coarse discretization ($\Delta t = 0.1$) and high CFL values ($\approx 8.0$), while audio tasks require fine temporal resolution ($\Delta t = 0.001 - 0.005$) and low CFL values ($\approx 0.001$) to preserve spectral structure. Task-specific propagation dynamics emerge: slow, sustained waves for audio enable temporal integration, while fast waves for visual tasks rapidly combine spatial features, paralleling neural oscillatory behavior across brain regions.

## 3.4 ABLATION STUDY

To evaluate the importance of BioRNN's structural components, we conducted an ablation study across three benchmark tasks. Each variant removed a specific architectural feature to isolate its functional role in frequency-resonant computation.

**Full Model**: Includes all components: wave propagation ($c > 0$), spatial coupling (gradient and divergence operators), the intermediate oscillation field $o$, and a structured 2D grid. Input modulation is enabled to drive frequency-selective activation.

**No Wave** ($c = 0$): Disables wave propagation but retains spatial coupling and the auxiliary field. Input modulation remains enabled to assess the necessity of wave propagation.

**No Spatial Coupling**: Removes gradient and divergence, isolating oscillators without spatial communication. Input modulation is retained to evaluate local frequency responses.

**Single Field**: Removes the auxiliary field $o$, which reduces the dynamics to a diffusion-like update on $p$ alone. Since this regime cannot support resonance, input modulation is also disabled, making this variant reflect the combined effect of removing $o$ and its associated modulation.

**Random Topology**: Preserves wave propagation and modulation but replaces the regular 2D grid with random connections.

Table 3: Ablation study on BioRNN. Test accuracy (%) is reported for the full model and several ablation variants where a specific component is removed. Values in parentheses indicate the accuracy drop compared to the full model. ESC50 results are reported as 5-fold means.

| Model | sMNIST | nsCIFAR-10 | ESC50-10 ms (5-fold) |
|---|---|---|---|
| **Full BioRNN** | **98.1** | **54.2** | **33.6** |
| No Wave | 94.98 (-3.12) | 43.32 (-10.88) | 28.75 (-4.85) |
| No Spatial | 95.75 (-2.35) | 44.60 (-9.60) | 29.90 (-3.70) |
| Single Field | 92.26 (-5.84) | 37.91 (-16.29) | 29.30 (-4.30) |
| Random Topology | 72.85 (-25.25) | 24.29 (-29.91) | 29.75 (-3.85) |

As shown in Table 3, each component reveals distinct functional roles. Removing wave propagation (No Wave) causes substantial drops on spectrally rich tasks (ESC50: -4.85%, nsCIFAR-10: -10.88%), indicating that wave dynamics are crucial for complex temporal processing. The auxiliary field $o$ proves essential across all tasks: without it (Single Field), performance degrades significantly, with the largest impact on nsCIFAR-10 (-16.29%), suggesting that coupling between $p$ and $o$ enables sophisticated spatiotemporal representations beyond simple diffusion. Interestingly, components show task-dependent importance: visual tasks are highly sensitive to spatial disruption (Random Topology: -25.25% sMNIST, -29.91% nsCIFAR-10), while ESC50 is more robust (-3.85%), indicating that structured spatial organization is critical for sequential visual processing, whereas audio benefits more from intrinsic frequency-selective properties. These results validate that BioRNN's effectiveness emerges from the synergistic integration of biologically motivated components.

## 4 CONCLUSION

This work introduced BioRNN, a recurrent architecture that unifies biologically inspired neuromodulatory preprocessing with physically grounded wave-equation dynamics. By embedding a mixed finite-difference scheme with learnable damping, BioRNN resolves the long-standing instability and incompatibility of physical wave equations with gradient-based training. Empirical evaluations across visual and auditory benchmarks demonstrated competitive performance with conventional gated RNNs and consistent improvements on frequency-rich auditory tasks. Ablation studies further showed that neither neuromodulation nor physical wave-equation dynamics alone suffices; their integration yields emergent spatiotemporal capabilities essential for resonance-based computation. Taken together, these findings establish BioRNN as a principled framework for recurrent models that are both biologically grounded and stably trainable, advancing the design of spatiotemporal architectures at the interface of physics and neuroscience.

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

APPENDIX

# A  RELATED WORK

## A.1  NEUROBIOLOGICAL UNDERPINNINGS

The mammalian brain exhibits a hierarchical organization in which specialized neuromodulatory nuclei, such as the locus coeruleus and basal forebrain, regulate cortical processing networks (Briand et al., 2007; McCormick et al., 2020). This two-level architecture separates contextual control from information processing. Neuromodulatory systems project broadly across cortical areas, delivering diffuse chemical signals that modulate the state of local circuits without directly encoding sensory inputs (Avery & Krichmar, 2017; Shine, 2023). This enables the same cortical circuitry to operate under distinct dynamical regimes, depending on the neuromodulatory context.

At a finer scale, neurons in the prefrontal cortex are arranged in a discrete, mosaic-like structure. Signal-carrying units communicate through intermediate neurons, allowing circuits to generate complex oscillatory dynamics from multi-dimensional interactions (Watakabe et al., 2023). These spatially organized neural populations produce rhythmic activity patterns driven by excitatory and inhibitory interactions, which propagate as wave-like dynamics and resonance phenomena across cortical areas (Lozano-Soldevilla & VanRullen, 2019; de Mooij-van Malsen et al., 2023). This biological architecture suggests a computational principle: adaptive control mechanisms (neuromodulation) should be separated from core processing dynamics (local computation).

The cortex's mosaic-like spatial organization enables wave-like signal propagation through local recurrent interactions (Bhattacharya et al., 2022; Sato, 2022). This inspires our bio-inspired RNN design, where an external modulation system dynamically regulates spatially organized recurrent layers. Our architecture is motivated by this biological evidence, incorporating separate modulation and processing components, neural grids, and wave-equation dynamics to support rich, context-dependent behavior.

## A.2  BIO-INSPIRED NEURAL ARCHITECTURES

Several neural architectures, such as TVAE (Keller & Welling, 2021), TDANN (Lee et al., 2020), and CORnet (Kubilius et al., 2019), incorporate spatial organization or recurrence to approximate cortical structure and hierarchical processing. These models emphasize topographic representations and feedback dynamics, providing valuable insights into the functional architecture of the brain.

Other approaches aim to capture rhythmic activity and wave-like behavior. CoRNN (Rusch & Mishra, 2021) employs coupled oscillators to model temporal dynamics, while the Neural Wave Machine (NWM)Keller & Welling (2023) uses a two-dimensional layout where local interactions produce traveling waves during visual tasks. The wRNNKeller et al. (2024) takes a more analytical approach, using a one-dimensional wave equation to simulate directional propagation and oscillatory patterns.

While these models each highlight different aspects of neural computation, they primarily focus on internal dynamics shaped by recurrent structure or local interactions. In contrast, our work centers on the interaction between an input modulator and a wave-based spatiotemporal neural network, motivated by neuroscience findings that emphasize the role of input-driven oscillations in shaping cortical activity. Specifically, we incorporate a modulatory preprocessing stage that transforms sensory input into oscillatory patterns and a coupling matrix grounded in physical wave principles. This design enables frequency-selective, task-dependent resonance, offering a biologically informed framework for modeling spatiotemporal dynamics in neural systems.

# B  STABILITY ANALYSIS OF BIORNN DISCRETIZATION

This appendix provides a rigorous analysis supporting the stability claims in Subsection **Dissipation-Driven Stability for BioRNN**. We follow the order of concepts as introduced in the main subsection: (1) frequency-domain interpretation of mixed finite differences, (2) derivation of the recurrent coupling matrix **A** according to mixed finite differences, and (3) comparison with central differences.

## B.1 Frequency-Domain Analysis of Mixed Difference Operators

This section provides the detailed proof of Lemma 1.

### B.1.1 Backward Difference (Gradient Operator)

The backward finite difference is defined as

$$\left(D_x^{\text{backward}} p\right)_{m,n} = \frac{p_{m,n} - p_{m-1,n}}{\Delta x}. \tag{B.1}$$

Applying the spatial translation theorem in Fourier space:

$$\mathcal{F}\left[\frac{p_{m,n} - p_{m-1,n}}{\Delta x}\right] = \frac{1 - e^{-i\xi_x \Delta x}}{\Delta x} \hat{p}(\xi_x, \xi_y). \tag{B.2}$$

Expanding the exponential:

$$e^{-i\xi_x \Delta x} = 1 - i\xi_x \Delta x - \frac{\xi_x^2 \Delta x^2}{2} + \mathcal{O}(\Delta x^3). \tag{B.3}$$

Substituting into equation B.2:

$$\tilde{\xi}_x^{\text{backward}} = \frac{1 - \left(1 - i\xi_x \Delta x - \frac{\xi_x^2 \Delta x^2}{2} + \mathcal{O}(\Delta x^3)\right)}{\Delta x}$$

$$= i\xi_x + \frac{\xi_x^2 \Delta x}{2} + \mathcal{O}(\Delta x^2). \tag{B.4}$$

Thus, the backward operator introduces a dissipative correction proportional to $+\xi_x^2 \Delta x / 2$.

### B.1.2 Forward Difference (Divergence Operator)

The forward finite difference is defined as

$$\left(D_x^{\text{forward}} o_x\right)_{m,n} = \frac{o_{x,m+1,n} - o_{x,m,n}}{\Delta x}. \tag{B.5}$$

Its Fourier transform is

$$\mathcal{F}\left[\frac{o_{x,m+1,n} - o_{x,m,n}}{\Delta x}\right] = \frac{e^{i\xi_x \Delta x} - 1}{\Delta x} \hat{o}_x(\xi_x, \xi_y). \tag{B.6}$$

Expanding the exponential:

$$e^{i\xi_x \Delta x} = 1 + i\xi_x \Delta x - \frac{\xi_x^2 \Delta x^2}{2} + \mathcal{O}(\Delta x^3). \tag{B.7}$$

Substituting into equation B.6:

$$\tilde{\xi}_x^{\text{forward}} = \frac{\left(1 + i\xi_x \Delta x - \frac{\xi_x^2 \Delta x^2}{2} + \mathcal{O}(\Delta x^3)\right) - 1}{\Delta x}$$

$$= i\xi_x - \frac{\xi_x^2 \Delta x}{2} + \mathcal{O}(\Delta x^2). \tag{B.8}$$

Thus, the forward operator introduces an anti-dissipative correction proportional to $-\xi_x^2 \Delta x / 2$.

## B.2 Full Derivation of recurrent coupling matrix A

This section supports **Theorem 2** in the main text. Set $\Delta x = \Delta y = 1$ for clarity.

### B.2.1 PHASE 1: VELOCITY UPDATE

$$\hat{o}_x^* = \hat{o}_x^n - \Delta t \cdot c \cdot \left( i\xi_x + \frac{\xi_x^2}{2} \right) \hat{p}^n \tag{B.9}$$

$$\hat{o}_y^* = \hat{o}_y^n - \Delta t \cdot c \cdot \left( i\xi_y + \frac{\xi_y^2}{2} \right) \hat{p}^n \tag{B.10}$$

$$\hat{p}^* = \hat{p}^n - \Delta t \cdot c \cdot \left[ \left( i\xi_x - \frac{\xi_x^2}{2} \right) \hat{o}_x^* + \left( i\xi_y - \frac{\xi_y^2}{2} \right) \hat{o}_y^* \right] \tag{B.11}$$

### B.2.2 COUPLING MATRIX FROM PHASE 1: $\mathbf{M}_{\text{VEL}}$

Combining equations equation B.9–equation B.11 yields the undamped velocity-phase coupling matrix:

$$\mathbf{M}_{\text{vel}} = \begin{bmatrix} M_{11}^{\text{vel}} & M_{12}^{\text{vel}} & M_{13}^{\text{vel}} \\ M_{21}^{\text{vel}} & M_{22}^{\text{vel}} & M_{23}^{\text{vel}} \\ M_{31}^{\text{vel}} & M_{32}^{\text{vel}} & M_{33}^{\text{vel}} \end{bmatrix}, \tag{B.12}$$

where

$$M_{11}^{\text{vel}} = 1 - (\Delta t \cdot c)^2 (\xi_x^2 + \xi_y^2) - \frac{(\Delta t \cdot c)^2}{4} (\xi_x^4 + \xi_y^4),$$

$$M_{12}^{\text{vel}} = -\Delta t \cdot c \left( i\xi_x - \frac{\xi_x^2}{2} \right), \quad M_{13}^{\text{vel}} = -\Delta t \cdot c \left( i\xi_y - \frac{\xi_y^2}{2} \right),$$

$$M_{21}^{\text{vel}} = -\Delta t \cdot c \left( i\xi_x + \frac{\xi_x^2}{2} \right), \quad M_{22}^{\text{vel}} = 1, \quad M_{23}^{\text{vel}} = 0,$$

$$M_{31}^{\text{vel}} = -\Delta t \cdot c \left( i\xi_y + \frac{\xi_y^2}{2} \right), \quad M_{32}^{\text{vel}} = 0, \quad M_{33}^{\text{vel}} = 1.$$

### B.2.3 PHASE 2: DAMPING UPDATE

The damping step contributes multiplicative attenuation via:

$$\mathbf{M}_{\text{damp}}^{-1} = \begin{pmatrix} \alpha & 0 & 0 \\ 0 & \beta & 0 \\ 0 & 0 & \beta \end{pmatrix}, \tag{B.13}$$

where $\alpha = \frac{1}{1+\Delta t \cdot k^p}$ and $\beta = \frac{1}{1+\Delta t \cdot k^o}$. To simplify the matrix inversion, $k^p$ and $k^o$ are set as diagonal matrices.

### B.2.4 FULL RECURRENT COUPLING MATRIX $\mathbf{A}$

Multiplying the damping matrix from Phase 2 with the velocity-phase matrix gives:

$$\mathbf{A} = \mathbf{M}_{\text{damp}}^{-1} \cdot \mathbf{M}_{\text{vel}} \tag{B.14}$$

This leads to the explicit form:

$$\mathbf{A} = \begin{bmatrix} A_{11} & A_{12} & A_{13} \\ A_{21} & A_{22} & A_{23} \\ A_{31} & A_{32} & A_{33} \end{bmatrix}, \tag{B.15}$$

where $\alpha = \frac{1}{1+\Delta t \cdot k^p} < 1$ and $\beta = \frac{1}{1+\Delta t \cdot k^o} < 1$, and

$$A_{11} = \alpha \left( 1 - (\Delta t \cdot c)^2 (\xi_x^2 + \xi_y^2) - \frac{(\Delta t \cdot c)^2}{4} (\xi_x^4 + \xi_y^4) \right),$$

$$A_{12} = -\alpha \Delta t \cdot c \left( i\xi_x - \frac{\xi_x^2}{2} \right), A_{13} = -\alpha \Delta t \cdot c \left( i\xi_y - \frac{\xi_y^2}{2} \right),$$

$$A_{21} = -\beta \Delta t \cdot c \left( i\xi_x + \frac{\xi_x^2}{2} \right), \quad A_{22} = \beta, \quad A_{23} = 0,$$

$$A_{31} = -\beta \Delta t \cdot c \left( i\xi_y + \frac{\xi_y^2}{2} \right), \quad A_{32} = 0, \quad A_{33} = \beta.$$

### B.3 COMPARISON WITH CENTRAL DIFFERENCES

This section supports **Theorem 1** in the main text.

#### B.3.1 CENTRAL DIFFERENCE EXPANSION

We consider a central difference discretization for both gradient and divergence:

$$\left. \frac{\partial p}{\partial x} \right|_{m,n} \approx \frac{p_{m+1,n} - p_{m-1,n}}{2\Delta x}, \tag{B.16}$$

$$\left. \frac{\partial p}{\partial y} \right|_{m,n} \approx \frac{p_{m,n+1} - p_{m,n-1}}{2\Delta y} \tag{B.17}$$

The Fourier transform yields:

$$\mathcal{F} \left[ \frac{p_{m+1,n} - p_{m-1,n}}{2\Delta x} \right] = \frac{e^{i\xi_x \Delta x} - e^{-i\xi_x \Delta x}}{2\Delta x} \hat{p}(\xi_x, \xi_y)$$
$$= \frac{i \sin(\xi_x \Delta x)}{\Delta x} \hat{p}(\xi_x, \xi_y) \tag{B.18}$$

Using the Taylor expansion $\sin(\xi_x \Delta x) = \xi_x \Delta x - \frac{(\xi_x \Delta x)^3}{6} + \mathcal{O}(\Delta x^5)$, we find:

$$\frac{i \sin(\xi_x \Delta x)}{\Delta x} \approx i\xi_x - i\frac{\xi_x^3 \Delta x^2}{6} + \mathcal{O}(\Delta x^4) \tag{B.19}$$

Central differences yield approximately $i\xi_x$ with no first-order dissipation, contrasting with the mixed difference scheme.

#### B.3.2 SPATIAL COUPLING MATRIX FOR CENTRAL SCHEME

By repeating the same operator splitting with central differences, the amplification matrix becomes:

$$\mathbf{A}_{\text{central}} = \begin{pmatrix} \alpha(1 - (\Delta t \cdot c)^2 |\boldsymbol{\xi}|^2) & -\alpha \Delta t \cdot c \cdot i\xi_x & -\alpha \Delta t \cdot c \cdot i\xi_y \\ -\beta \Delta t \cdot c \cdot i\xi_x & \beta & 0 \\ -\beta \Delta t \cdot c \cdot i\xi_y & 0 & \beta \end{pmatrix} \tag{B.20}$$

This lacks the crucial $\pm \xi^2/2$ terms found in the mixed scheme.

#### B.3.3 HIGH-FREQUENCY MODE STABILITY

We now analyze the $A_{11}$ term for high-frequency modes $|\xi| \to \pi$.

**Mixed Difference:**

$$A_{11}^{\text{mixed}} = \alpha - \alpha(\Delta t \cdot c)^2 \pi^2 \left( 1 + \frac{\pi^2}{4} \right) \tag{B.21}$$

**Central Difference:**

$$A_{11}^{\text{central}} = \alpha - \alpha(\Delta t \cdot c)^2 \pi^2 \tag{B.22}$$

Since $\pi^2/4 \approx 2.47$, the mixed scheme ensures:

$$A_{11}^{\text{mixed}} < A_{11}^{\text{central}} \quad \text{for high } |\xi| \tag{B.23}$$

Mixed differences offer superior suppression of high-frequency modes compared with central differences, enabling robust operation under supercritical CFL.

## C   HYPERPARAMETERS AND ENVIRONMENT

We evaluate our model on three benchmark datasets: sMNIST, nsCIFAR10, and ESC50. Dataset splits are summarized in Table C.1. Final results reported in the main text were obtained using the optimal hyperparameters identified in Table D.1.

All models were implemented in PyTorch and trained using the Adam optimizer for 120 epochs. To ensure reproducibility, all experiments, including task comparisons, grid search, and ablation studies, were conducted with a fixed random seed (`torch.manual_seed(42)`). Accuracy (percent) was used as the evaluation metric across all datasets. All experiments were performed on a single NVIDIA GeForce RTX 4090 GPU with 24 GB of memory.

Table C.1: Training, validation, and test splits for the three datasets used in our experiments.

| Dataset | Training Set | Validation Set | Test Set |
|---|---|---|---|
| sMNIST | 57,000 | 3,000 | 10,000 |
| nsCIFAR10 | 47,000 | 3,000 | 10,000 |
| ESC50 | 1,600 | 200 | 200 |

For the image domain, we employ the *noisy CIFAR-10* (nsCIFAR10) benchmark, which is derived from the standard CIFAR-10 dataset containing 60,000 color images across 10 classes. To simulate label imperfections commonly encountered in real-world scenarios, a fraction of the training labels are intentionally corrupted according to predefined noise schemes. In our experiments, we adopt the commonly used symmetric noise setting, where labels are flipped uniformly at random to other classes, while maintaining clean labels in the validation and test sets.

The results are reported on the ESC-50 dataset (Table C.2) using its official five-fold cross-validation protocol. The ESC-50 corpus consists of 2000 environmental audio recordings evenly distributed over 50 classes. Following the standard evaluation procedure, the dataset is partitioned into five predefined folds, where each fold is used once as the test set while the remaining four folds serve for training. The table reports the classification accuracy (%) obtained on each individual fold (**Fold 1**–**Fold 5**), while the column **Overall** summarizes the mean and standard deviation of the accuracy across the five folds.

Table C.2: ESC-50 classification accuracy (%) for each fold. Overall reports the mean $\pm$ standard deviation across the 5 folds.

| Group | Model | Fold 1 | Fold 2 | Fold 3 | Fold 4 | Fold 5 | Overall |
|---|---|---|---|---|---|---|---|
| *Base* | LSTM | 32.0 | 27.8 | 33.8 | 37.0 | 31.3 | 32.4 $\pm$ 3.4 |
| | GRU | 42.0 | 41.8 | 43.8 | 43.8 | 39.8 | 42.2 $\pm$ 1.7 |
| *coRNN* | coRNN (no mod) | 4.00 | 16.25 | 16.75 | 3.25 | 12.00 | 10.45 $\pm$ 6.50 |
| | coRNN (mod.) | 3.50 | 9.50 | 10.75 | 2.75 | 4.00 | 6.10 $\pm$ 3.73 |
| *NWM* | NWM (no mod) | 4.3 | 3.3 | 1.8 | 6.3 | 4.3 | 4.0 $\pm$ 1.6 |
| | NWM (mod.) | 4.3 | 3.8 | 5.8 | 9.8 | 4.8 | 5.7 $\pm$ 2.4 |
| *wRNN* | wRNN (no mod) | 14.5 | 14.5 | 21.5 | 16.0 | 17.3 | 16.8 $\pm$ 2.9 |
| | wRNN (mod.) | 13.8 | 14.5 | 18.5 | 22.3 | 17.0 | 17.2 $\pm$ 3.4 |
| *BioRNN* | BioRNN (no mod) | 29.5 | 29.3 | 34.5 | 37.3 | 33.0 | 32.7 $\pm$ 3.4 |
| | BioRNN (mod.) | 29.3 | 31.0 | 35.0 | 40.3 | 32.5 | 33.6 $\pm$ 4.3 |

To further investigate the role of the modulator and confirm that its parameters are indeed learned rather than manually designed, we visualize the distribution of its frequency parameters ($f$) before and after training. For the purpose of this visualization, we report the learned parameters from a representative single-fold training run (Fold 1) on the ESC-50 dataset. As shown in Figure C.1, the model is initialized with a generic linear distribution of frequencies (gray dashed line). After end-to-end training, the learned frequencies (red solid line) deviate from this initialization, exhibiting fine-grained local adjustments. This "micro-tuning" demonstrates that the modulator is actively learning to adapt the oscillatory drive to the specific spectral characteristics of the dataset, effectively performing impedance matching between the input signals and the resonant modes of the BioRNN grid.

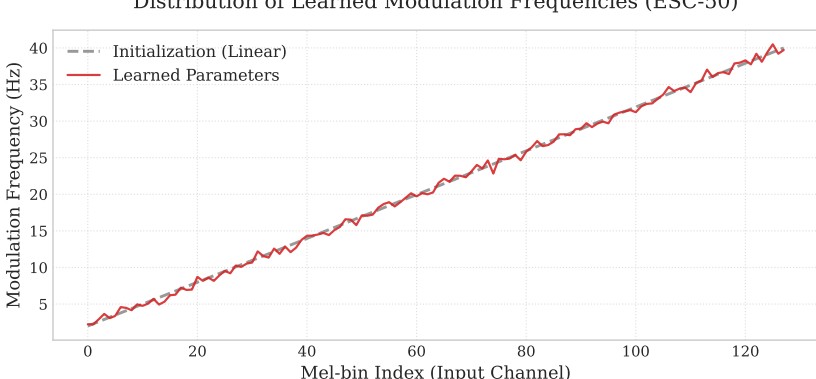

Figure C.1: **Frequency modulation visualization (ESC-50).** Learned frequency parameters ($f$) before (**gray dashed**, linear init) and after (**red solid**) training, shown for a representative single fold. The fine-grained local deviations ($\pm1$–2 Hz) demonstrate that the modulator acts as a precision tuner, performing "impedance matching" between specific input features and the resonant dynamics of the BioRNN wave grid.

## D  DETAILED ANOVA RESULTS

To better understand how each parameter contributes to model performance across tasks, we performed both marginal (one-way) and multi-way ANOVA analyses. The results are summarized in Tables D.1 and D.2, and discussed in the following paragraphs.

We first present a one-way ANOVA where each parameter was varied independently to evaluate its marginal influence on the model's accuracy. As shown in Table D.1, the time step size $\Delta t$ consistently dominates across all tasks, with marginal effects exceeding 90% in sMNIST and Noisy CIFAR, and remaining substantial in ESC50-100 ms (67.94%) and ESC50-10 ms (78.25%). The relative impact of $c$, grid size, and $k$ is much smaller across tasks. Interestingly, the influence of $c$ becomes more pronounced in complex tasks, such as ESC50-10 ms, where it reaches 6.53%. These results suggest that $\Delta t$ is the most sensitive parameter across all regimes, controlling temporal resolution and stability, while other parameters only exert secondary effects.

To further account for interactions between parameters, we then performed a multi-way ANOVA. Table D.2 reports the explained variance ($\eta^2$) for each main effect and pairwise interaction term. While $\Delta t$ remains the dominant main effect, explaining 90.87% of variance in sMNIST and 95.48% in Noisy CIFAR, its contribution in more complex tasks like ESC50-100 ms drops to 50.56%, with a notable portion of variance captured by interactions such as $\Delta t \times$ grid (4.81%) and $\Delta t \times c$ (3.21%). In ESC50-10 ms, a similar trend is observed: while $\Delta t$ still explains 76.64% of the variance, interaction effects are non-negligible, particularly with $c$ (4.07%).

The small values of $\eta^2$ for $k$ in all tasks, as well as the weak interactions involving $k$, indicate that this parameter has limited influence on performance in the tested configurations. Grid size and $c$ show increasing importance as task complexity and sequence length increase, reflecting their roles in controlling spatial resolution and wave propagation.

Although the marginal effect analysis and multi-way ANOVA largely agree on the dominant role of $\Delta t$, some discrepancies are observed in the magnitude of their contributions. For example, in ESC50-100 ms, the marginal effect of $\Delta t$ is 67.94%, while its $\eta^2$ is 50.56%. This difference arises because marginal effects are computed by varying each parameter in isolation, potentially overestimating its importance when interdependencies are present. In contrast, $\eta^2$ partitions variance orthogonally across all factors and their interactions, leading to a more conservative estimate. As such, the marginal and multi-way analyses provide complementary perspectives: the former highlights sensitivity, while the latter reflects statistical attribution.

Table D.1: Comparison of Parameter Importance and Optimal Values Across Four Tasks (sMNIST, noisy CIFAR-10, ESC50 at 100 ms, and ESC50 at 10 ms). For each task, the table lists model parameters in descending order of importance (ranked by their marginal effect on performance), along with the percentage of explained variance (marginal effect) attributed to each parameter and its corresponding optimal value. The ranking highlights which parameters most strongly influence performance within each task.

| Task | Param | Effect (%) | Opt. Val. | Rank |
|---|---|---|---|---|
| sMNIST | $\Delta t$ | **95.72** | 0.1 | 1 |
| | $c$ | 15.14 | 80 | 2 |
| | grid | 10.72 | 64 | 3 |
| | $k$ | 0.74 | 0.05 | 4 |
| Noisy CIFAR | $\Delta t$ | **95.55** | 0.1 | 1 |
| | $c$ | 0.44 | 80 | 2 |
| | grid | 0.42 | 64 | 3 |
| | $k$ | 0.02 | 0.1 | 4 |
| ESC50-100 ms | $\Delta t$ | **67.94** | 0.005 | 1 |
| | grid | 15.29 | 32 | 2 |
| | $c$ | 5.26 | 1 | 3 |
| | $k$ | 0.52 | 0.1 | 4 |
| ESC50-10 ms | $\Delta t$ | **78.25** | 0.001 | 1 |
| | $c$ | 6.53 | 1 | 2 |
| | grid | 4.17 | 32 | 3 |
| | $k$ | 0.21 | 0.01 | 4 |

Table D.2: Explained Variance ($\eta^2$, in %) of Model Parameters and Their Interactions Across Tasks. Each row corresponds to a main parameter or a pairwise interaction (e.g., $\Delta t \times c$), and each column represents a specific task: sMNIST, noisy CIFAR-10 (nsCIFAR10), ESC50 with 100 ms frames, and ESC50 with 10 ms frames. The values indicate the proportion of total performance variance (in percentage) attributable to each parameter or interaction term, based on an ANOVA-style decomposition. Higher $\eta^2$ indicates a greater influence on model performance for that task.

| Source | sMNIST | nsCIFAR10 | ESC50 (100 ms) | ESC50 (10 ms) |
|---|---|---|---|---|
| **Main Effects** | | | | |
| $\Delta t$ | **90.87** | **95.48** | **50.56** | **76.64** |
| c | 5.21 | 0.44 | 5.53 | 7.61 |
| grid | 0.78 | 0.43 | 7.92 | 2.20 |
| k | 0.45 | 0.02 | 0.19 | 0.04 |
| | | | | |
| **Two-way** | | | | |
| $\Delta t \times c$ | 2.07 | 1.36 | 3.21 | 4.07 |
| $\Delta t \times$ grid | 1.18 | 0.52 | 4.81 | 2.43 |
| $\Delta t \times k$ | 0.89 | 0.35 | 1.38 | 1.02 |
| c $\times$ grid | — | — | 1.83 | 0.59 |
| c $\times$ k | — | — | 0.70 | 0.42 |
| grid $\times$ k | 0.16 | 0.08 | 0.47 | 0.30 |

# E    LLM USAGE DECLARATION

This manuscript benefited from the use of a large language model (LLM) to check grammar, spelling, and typographical consistency. The authors are fully responsible for the content, claims, and conclusions presented in the paper.

