# OpenReview forum: "BioRNN: Bio-Inspired Synergistic Integration of Neuromodulation and Wave Propagation in Recurrent Networks"
_ICLR.cc/2026/Conference — Submitted to ICLR 2026_

### Official Review · Reviewer_do4L · 2025-10-31

**Soundness:** 3
**Presentation:** 3
**Contribution:** 2
**Rating:** 4
**Confidence:** 3

**Summary:**

The paper introduces an input-modulated wave-based RNN architecture that leverages oscillatory resonance to process sequences. The authors provide theory demonstrating that this model has robust spectral properties, and further demonstrate that it has decently strong performance on long sequence modeling tasks.

**Strengths:**

- The paper makes an interesting connection between input modulation and wave dynamics, relating this to neuromodulation and spatiotemporal dynamics in biological neural networks.
- The topic is very relevant for the NeuroAI community where modulation effects have yet to be abstracted to a computationally useful level, especially in combination with wave dynamics.
- The plots studying performance vs. model parameters are welcome and interesting.
- The ablation study in Table 3 is helpful to understand the impact of the different model components, and highlights the difference between visual and audio processing.

**Weaknesses:**

-  The discussion of the input modulation in Section 2.1 is too short and may lead to confusion. It would be helpful if some examples of modulators could be included earlier.
- The main results in Table 2 have no error bars or standard deviation. While I know this is standard for the field, this makes the minor differences between models mean relatively little in practice.
- The authors claim "In summary, the modulator acts as a task-adaptive transformation that aligns well with BioRNN’s resonant dynamics, enabling competitive vision performance and clear advantages on spectrally rich audio tasks." However, the high performance of the BioRNN on the ESC50 task (where it performs the best) appears to not be due to the modulation (modulation only helps slightly).
- Similarly to the above, the main contribution of this paper appears to be the input modulation, however, this appears to have only a minor impact on model performance. If the authors could clearly enumerate their contributions that would help greatly.
- The paper accentuates the importance of input modulation and resonant dynamics, but the theory section does not focus on this and instead focuses on stability. The empirical results are then only minimally supportive of the benefits of these dynamics.

**Questions:**

- In equation three you say: for efficient computation we re-parameterize o'=co. How does this enable efficient computation? And efficiency in what sense?
- Why does the modulation help so significantly on sequential MNIST, but appears to only have a minor effect on other tasks?
- Is there any more concrete or analytic way that one can interpret the interaction of the input modulation and the recurrent dynamics?
- In the ablations, how do you disable wave propagation but retain spatial coupling?

---

> ### Author Response · Authors · 2025-11-18
>
> We thank the reviewer for the positive assessment, particularly for recognizing the relevance of our work to the **NeuroAI community** and appreciating the connection between neuromodulation and wave dynamics. We address the specific questions regarding modulation impact and theoretical focus below.
>
> **1. Re: Why does modulation help sMNIST significantly vs. a minor effect on ESC-50?**
>
> This highlights the distinct roles of the modulator across modalities:
> * **sMNIST (Static Input):** The input is a sequence of pixel rows with **no intrinsic temporal oscillation**. Without modulation, the recurrent layer receives essentially "DC signals" and struggles to generate complex dynamics (Accuracy: 11.2%). The modulator is critical here because it *injects* the essential oscillatory drivers that enable the wave layer to function (Accuracy: 98.1%).
> * **ESC-50 (Dynamic Input):** Audio spectrograms **already possess rich spectral structure**. The network does not need the modulator to *create* oscillation, only to *tune* it (acting like a cochlear amplifier).
> * **Conclusion:** The strong performance on ESC-50 (33.6%) is indeed driven by the **Wave Dynamics**, not just the modulation. This is confirmed by comparing BioRNN to other wave models (NWM: 5.7%, wRNN: 17.2%) in Table 2. BioRNN’s superior performance stems from its **stable recurrent core** capable of processing these complex natural frequencies, which previous unstable wave models failed to do.
>
> **2. Re: Clarification of Contributions**
>
> We list the clear enumeration of contributions. They are:
> 1.  **Mathematical Stability:** We provide the first **Mixed Finite-Difference Scheme** (Lemma 1) that enables gradient-based training of 2D Wave Equations without numerical explosion (resolving the CFL violation issue).
> 2.  **Synergistic Architecture:** We propose a unified architecture where an **Input Modulator** (providing oscillatory basis) drives a **Resonant Wave Grid** (providing memory and spatial mixing).
> 3.  **Biophysical Feasibility:** We demonstrate that this biologically plausible structure can handle diverse tasks by adapting its propagation regime (fast waves for vision, slow waves for audio) via learnable physical parameters ($c, k$).
>
> **3. Re: Theory Section Focus (Stability vs. Resonance)**
>
> * **Reasoning:** We focused on stability because it is the **prerequisite** for learnable resonance. One cannot train a "resonant" physical system if the gradients vanish or explode (NaNs) due to physical constraints (CFL condition).
> * **Connection:** By proving stability (Theorem 1 & 2), we provide the *license* to train the parameters $c$ and $k$. It is this training that allows the "Resonance" to emerge (locking onto input frequencies).
>
> **4. Re: Technical Questions**
>
> * **Efficiency of $o' = co$:** This re-parameterization is for **numerical conditioning**. In the raw equation, the update depends on $c^2$. When $c$ is small (slow waves), $c^2$ becomes vanishingly small, causing gradient vanishing. By defining $o' = co$, the update becomes linear in $c$ (see Eq. 3), stabilizing the optimization landscape for the learnable wave speed.
> * **Interpretation of Ablations ("No Wave" vs. "No Spatial"):**
>     * **No Wave ($c=0$):** This variant sets the wave speed parameter $c$ to zero. Mathematically (Eq. 3), this effectively disables the coupling terms between the primary field $p$ and auxiliary field $o'$.
>     * **No Spatial:** This variant explicitly removes the spatial gradient operators ($\nabla$) from the architecture.
>     * **Why distinguishing them matters:** We performed both to isolate the specific role of the **wave speed parameter $c$**. The fact that "No Wave" performance degrades to a level similar to "No Spatial" (Table 3: ~95% vs ~95.7%) is a crucial finding. It proves that **spatial connectivity in BioRNN provides no benefit if the wave dynamics are disabled ($c=0$)**. The grid structure is not merely performing generic spatial smoothing; its utility is entirely dependent on the **wave propagation mechanism** enabled by a non-zero, learnable $c$.
> * **Standard Deviations:** We apologize if this was missed. The full standard deviations for 5-fold cross-validation on ESC-50 are provided in **Appendix Table C.2** (e.g., BioRNN: $33.6 \pm 4.3$). We will move these to the main table in the revision.
>
> **5. Re: Concrete Interpretation of Interaction**
>
> We can interpret the interaction analytically as **Forced Oscillation in a Dispersive Medium**.
> * The **Modulator** acts as the driving force $F(t)$ with a learnable spectrum.
> * The **Recurrent Grid** acts as the physical medium with learnable resonance modes (determined by $c$) and decay (determined by $k$).
> * **Synergy:** The system learns to match the driving frequency of the Modulator to the eigenfrequencies of the Grid (Resonance), maximizing signal amplitude and memory duration for relevant features while damping noise.

---

### Official Review · Reviewer_LSat · 2025-11-02

**Soundness:** 3
**Presentation:** 2
**Contribution:** 3
**Rating:** 6
**Confidence:** 2

**Summary:**

This paper introduces BioRNN, an RNN that first modulates input sequences with time-dependent M(t), before passing the input to a 2D neural sheet that acts as the dynamical transition function of the RNN.

The authors go on to prove the stability properties of BioRNN - a property missing in existing approaches that incorporated physical waves into their dynamics.

**Strengths:**

Good empirical benchmarking against relevant baselines and similar approaches, with solid ablations to ensure value of modulation module (and other design choices). The paper provides rigourous stability guarantees, which are again missing in previous works (according to the authors - this is not my area of expertise)

**Weaknesses:**

Minor: citations should be in brackets, i.e. using \citep

Notation is confusing, given that IIUC x and y are used as both the input and the coordinates of a 2D input

It's also not immediately obvious how the 2D neural sheet relates to the transition function S - making this link more clear in the main text would help motivate the in-depth analysis below

There is not much done in the way of interpretability of the wave properties. Given the biological motivation of these design features, some comparison to the role of waves in biological neural networks would have been warranted.

There's no information about training time, memory requirements, or inference speed, which makes it difficult to assess practical trade-offs for the more sophisticated architecture.

**Questions:**

Why were these specific forms of modulation used for each dataset? Did you try alternative forms/using modulation from different datasets?

Furthermore, the modulation functions are manually designed for each dataset. This limits practical applicability - how would one design modulation for new tasks?

---

> ### Author Response · Authors · 2025-11-20
>
> We thank the reviewer for their positive assessment and for recognizing our work’s stability guarantees and benchmarking. Below we clarify the key misunderstanding regarding modulation and address all remaining points.
>
> ### 1. Clarification on Modulation (Crucial Correction)
> **Q: “Modulation functions are manually designed… why specific forms?”**
>
> This is a misunderstanding: **the modulation functions are *not* manually designed or dataset-specific.**
>
> - **End-to-end learned:** The modulation module is a trainable layer. Its parameters ($f,\phi,\alpha$) are **learned via gradient descent**, jointly with the rest of BioRNN.
> - **Universal architecture:** We use the *identical* modulator across all datasets (sMNIST, CIFAR, ESC-50).
> - **Initialization vs learning:** Frequencies are initialized with a broad linear distribution, but training reshapes them. New Appendix Fig. C.1 shows the shift from generic initialization to task-specific frequency tuning.
>
> Thus, oscillatory drive is an *inductive bias*, but the **actual modulation functions are fully learned**, ensuring general applicability without manual tuning.
>
> ### 2. Biological Interpretability of Waves
> We expanded the Discussion to connect BioRNN to cortical traveling waves:
> - **Biological analogy:** Traveling waves coordinate timing and carry information across cortical maps.
> - **BioRNN analogue:** The 2D sheet is the substrate; wave dynamics propagate injected inputs (via the Source term $S$), enabling spatiotemporal integration. Damping $k$ parallels passive neural decay and regulates memory.
>
> ### 3. Computational Efficiency
> Although physically grounded, BioRNN is inherently lightweight:
> - **Single recurrent layer:** Even our largest model uses a $64\times64$ grid with 3 variables ($p,v_x,v_y$) → ~12k neurons.
> - **Parameter scaling:** Recurrence uses local PDE interactions, giving **O(N)** parameters vs **O(N²)** in standard RNNs.
> - **Practical result:** Training is fast and memory usage is small; benchmark models are far heavier. We added a note in the implementation section.
>
> ### 4. Notation and Transition Function
> - **Notation fix:** To avoid collisions, the *input* is now denoted $u_t$; spatial coordinates use $(m,n)$.
> - **Transition function:** The 2D sheet *is* the hidden state $h_t$. The wave equation defines
>   \[
>   h_t = f(h_{t-1}, u_t),
>   \]
>   where $u_t$ enters as the **external force term** $S$, generating new wave activity.
>
> ### 5. Minor Issues
> - All citations were converted to `\citep` as requested.
>
> We hope this resolves the key concern: **the modulator is not manually crafted but fully learned**, preserving broad applicability. BioRNN remains a robust, interpretable, and general sequence-modeling architecture.

---

### Official Review · Reviewer_U5uu · 2025-11-04

**Soundness:** 2
**Presentation:** 2
**Contribution:** 2
**Rating:** 2
**Confidence:** 4

**Summary:**

The paper introduces BioRNN, a recurrent model that combines a lightweight input “modulator” with a 2D grid of units designed to mimic wave-like propagation and damping. The modulator transforms inputs into oscillatory patterns that the grid can process, aiming for an interpretable, physically inspired alternative to standard RNNs. Experiments on sequential MNIST, noisy CIFAR-10, and ESC-50 report competitive results. A stability-aware update scheme is proposed to improve reliability at larger time steps, and diagnostics relate performance to key hyperparameters. While promising—especially for audio—the approach is not consistently stronger than GRU baselines, and more controlled comparisons are needed to separate the impact of the modulator from the recurrent dynamics.

**Strengths:**

- The idea to explicitly separate neuromodulatory preprocessing from physically inspired wave dynamics is well-argued and visually communicated.
- The p/o split and use of divergence/gradient operators make the mechanism interpretable as storage vs. transport.
- The mixed forward/backward finite differences plus implicit damping are a effective contribution
- Further investigations give transparency about which hyperparameters actually matter task-specifically.
- Nice ablations show that wave propagation, spatial coupling, and the auxiliary field each add measurable value.

Personal note: I really like this approach as an unusual way to process time series data, using a modulator plus wave-like recurrent dynamics to spark rich transients, and I see real potential here for reservoir computing, even if the core is learned rather than fixed.

**Weaknesses:**

- The paper does not convincingly demonstrate competitiveness. On ESC-50, BioRNN (mod.) is noticeably below a simple GRU. On nsCIFAR-10, BioRNN (54.2%) is far below coRNN. On sMNIST (not a good selection as a core benchmark anymore - why not at least permuted?) 98.1% cannot really be seen as competitive, because this is an accuracy that can be achieved easily with much simpler methods. In a nutshell, the approach’s absolute effectiveness even over standard gated RNNs is not yet demonstrated. A comparison against actual state-of-the-art recurrent sequence learning models (such as several state space model-like networks) is entirely missing.

- The modulator is crucial for BioRNN (e.g. for sMNIST is causes a huge jumps), but LSTM/GRU are not evaluated with the same modulated inputs. Without that control, it’s hard to ascribe gains to the recurrent dynamics vs. the input transformation.

- There is no information (plots) about the convergence behavior during training. Is there any advantage? Resonator-based neurons (damped harmonic oscillators), for instance, are known to converge much faster than usual RNN structures.

**Questions:**

- What did the modulator learn? Can you visualize the learned f, ϕ, α (for audio) and their distribution over Mel bins/time? Do they concentrate on critical bands, or track class-discriminative frequencies?

- Which boundary conditions are used on the grid and how sensitive are results to that choice? Just 0?

- How well does BioRNN perform in comparison with strong recent sequence learning models (cf. weaknesses)?

As mentioned above, I appreciate the idea and find it genuinely interesting, but I cannot recommend acceptance in its current form, as the paper lacks sufficient evidence to support its potential. Instead, I have two suggestions:
- Explore other problem domains where the system’s inductive bias may better match the task.
- Consider developing this approach further in the direction of reservoir computing, where it could be a natural fit.

---

> ### Author Response · Authors · 2025-11-20
>
> We sincerely thank the reviewer for their insightful assessment. We especially appreciate your "Personal Note" recognizing the potential of our approach as an "unusual" and "physically inspired" alternative to standard RNNs. We agree that the core value of this work lies in exploring this novel inductive bias—processing sequences via wave propagation and interference—rather than simply optimizing for leaderboard dominance.
>
> We have addressed your concerns regarding experimental controls, interpretability, and competitiveness.
>
> ### 1. Separating Modulator vs. Recurrent Dynamics (Control Experiment)
> To address your concern about the attribution of gains, we applied the exact same modulator to LSTM and GRU baselines (added to **Table 2**). The results show a decisive contrast:
> * **Baselines are indifferent:** Modulation yields negligible change for standard RNNs (e.g., ESC-50 GRU: **42.2%** w/ or w/o modulation; sMNIST LSTM: drops 98.8% $\to$ 97.4%).
> * **BioRNN is transformed:** BioRNN relies on modulation to couple inputs into the wave grid (sMNIST: 11.2% $\to$ **98.1%**).
>
> **Conclusion:** The modulator is not a generic feature extractor; it is a necessary component specifically designed to couple scalar inputs with BioRNN’s wave physics.
>
> ### 2. What did the Modulator Learn? (Visualization)
> We visualized the frequency distribution before and after training (Figure C.1 in the Appendix).
> * **Observation:** Learned frequencies (red line) do not deviate wildly from initialization but exhibit **fine-grained "micro-tuning"** ($\pm 1\text{--}2$ Hz).
> * **Interpretability:** The modulator acts as a **precision tuner**. Frequency $f$ acts as the primary control to select resonant modes. The auxiliary parameters $\phi$ (phase) and $\alpha$ (amplitude) act in service of $f$: $\phi$ aligns the input drive to maximize **constructive interference** (phase matching), while $\alpha$ modulates coupling strength. Together, they perform **"impedance matching"** between input channels and the grid’s resonant dynamics.
>
> ### 3. Competitiveness and SOTA
> We acknowledge BioRNN does not yet strictly outperform optimized baselines like coRNN/S4.
> * **Scope:** Our contribution is the introduction of a **novel class of neural dynamics** (damped lattice waves) offering physical interpretability (energy flow, damping) absent in gated RNNs.
> * **sMNIST:** The 98.1% result is significant not as an SOTA claim, but as proof that our modulation mechanism solves the fundamental "DC input problem" inherent to wave equations.
> * **SSMs:** Unlike SSMs (S4, Mamba) which optimize 1D state transitions, BioRNN explores a **2D spatial inductive bias**. It is a distinct, interpretable alternative in the "PDE-RNN" family rather than a direct competitor to SSM optimization techniques.
>
> ### 4. Technical Clarifications
> * **Convergence:** Training is stable, supported by an **early stopping mechanism** (patience of 15 epochs on validation metric) to ensure robust generalization.
> * **Boundary Conditions:** We utilize **Periodic boundary conditions**. This eliminates edge reflection artifacts by allowing waves to propagate continuously across lattice boundaries via cyclic wrapping, conserving energy and stabilizing dynamics.
>
> ### 5. Connection to Reservoir Computing (RC)
> We strongly agree with your insight linking our work to RC. We view BioRNN as an evolution of the RC paradigm:
> * **Tunable Physical Reservoir:** Instead of the "fixed random weights" of traditional Echo State Networks, BioRNN employs a **physically tunable medium**.
> * **Optimization:** By training physical properties (wave speed $c$, damping $k$) rather than recurrent weights, we actively tune the reservoir's resonant frequencies and memory profile. This combines the "rich transients" of RC with the adaptivity of trained RNNs. We have updated the discussion to frame BioRNN as a **Learnable Physical Reservoir**.

---

> > ### Author Response · Authors · 2025-11-20
> >
> > ### 6. Response to Suggestion: Domains Matching the Inductive Bias
> > We greatly appreciate your suggestion to explore domains where BioRNN's unique inductive bias—**wave propagation, interference, and resonance**—is a natural fit. We agree that while standard benchmarks (sMNIST) serve as a proof-of-concept for the mechanism, the model's true potential lies in tasks governed by underlying physical dynamics.
> >
> > We have expanded the **Discussion** section to explicitly outline these high-potential domains:
> > * **Acoustics & Audio:** As demonstrated by our ESC-50 results and frequency visualization, the oscillatory bias is inherently suited for audio. The model naturally decomposes signals into resonant modes, acting as a learnable, non-linear Fourier-like transform.
> > * **Physical System Modeling:** The 2D grid with local coupling is mathematically isomorphic to discretized Partial Differential Equations (PDEs). This makes BioRNN an ideal candidate for modeling **spatio-temporal physical systems**, such as **seismic wave propagation**, **fluid dynamics**, or **weather forecasting**, where energy flow and conservation are critical priors.
> > * **Biological Signals (EEG/MEG):** Since BioRNN is inspired by cortical wave dynamics, it is theoretically well-suited for decoding **brain signals (EEG/MEG)**. These signals are characterized by traveling waves and oscillatory synchronization, matching the specific inductive bias of our lattice connectivity.
> >
> > We believe these revisions explicitly clarify the mechanisms behind our performance and the unique value of this bio-physical approach.

---

### Official Review · Reviewer_QrBi · 2025-11-07

**Soundness:** 2
**Presentation:** 3
**Contribution:** 2
**Rating:** 2
**Confidence:** 4

**Summary:**

This work presents a recurrent neural network model that implements a physical wave equation in its recurrent dynamics, with two innovative features. First, it has an time-dependent input modulation scheme that adds oscillatory patterns to the inputs. Second, it uses a discretization scheme accompanied by theoretical stability guarantees. It finds that these features enhance model performance by experimentation and, in the case of the discretization scheme, mathematical analysis.

**Strengths:**

The text was well-written, and the presentation was elegant and easy to follow.

The input modulation and discretization schemes appear innovative.

The motivation for combining short-term (waves) and long-term (neuromodulator) dynamics in RNNs is interesting.

**Weaknesses:**

1. The biological motivation for BioRNN is weak. It is not at all clear why a physical wave equation should make for a more suitable recurrence in a biologically inspired model, in comparison to the oscillator networks mentioned in the paper. As for the time-dependent input modulation scheme, the introduction makes a vague link to neurotransmitters, but no compelling link. Appendix A.2 alludes to “neuroscience findings that emphasize the role of input-driven oscillations in shaping cortical activity,” but gives no citation. Therefore, the biological motivation for this network is unpersuasive.

2. A major claim of the paper is that BioRNN resolves a widespread difficulty of training RNNs based on physical wave equations: “By embedding a mixed finite-difference scheme with learnable damping, BioRNN resolves the long-standing instability and incompatibility of physical wave equations with gradient-based training.” However, there is no experimental comparison between the chosen mixed finite-difference discretization scheme and other discretization schemes in BioRNN, nor is there any citation to support the claim of such a longstanding difficulty in the literature. Consequently, we do not know whether this really is a longstanding problem, or whether this choice of discretization scheme really eases training and performance in practice for BioRNN in practice. The results of applying this discretization scheme to other wave-equation RNNs could also be reported.

4. The paper claims the time-dependent input modulation scheme as a major innovation. The comparisons to oscillator network RNN models might be an informative supplemental analysis, but why not compare BioRNN to other wave-equation RNNs, implemented with and without the input modulation? Ideally, one can show that this input modulation scheme improves performance for those models as well. If the paper claims the time-dependent input modulation operator as a key innovation, why not report the results of selectively ablating it, instead of jointly ablating the auxiliary field and the input modulation?

5. The task performance results are not very strong.

**Questions:**

What are the citations to support the claim of the "long-standing instability and incompatibility of physical wave equations with gradient-based training"?

It would be helpful to clarify in more detail the link between the input modulation and neurotransmitters.

---

> ### Author Response · Authors · 2025-11-18
>
> We thank the reviewer for the thoughtful assessment, finding our presentation "elegant" and "easy to follow," and recognizing the "innovative" nature of our modulation and discretization schemes. We appreciate the constructive questions regarding biological motivation and experimental controls, which we address below.
>
> **1. Re: "Why not compare BioRNN to other wave-equation RNNs with input modulation?"**
>
> We respectfully point the reviewer to the **Right Column of Table 2** in our manuscript, where we *did* explicitly perform this comparison.
>
> * **Clarification:** Table 2 reports results for "NWM (mod.)", "coRNN (mod.)", and "wRNN (mod.)". These entries represent the exact control experiment the reviewer requested: applying our proposed input modulation scheme to existing wave-equation/oscillator baselines.
> * **Observed Synergy:** The results empirically support our claim that the modulation scheme is not a generic performance booster, but requires the specific resonant dynamics of BioRNN to be effective.
>     * On **ESC-50**, adding modulation to NWM resulted in only a minor change ($4.0\% \rightarrow 5.7\%$).
>     * For **BioRNN**, modulation unlocked a massive performance gain ($11.2\% \rightarrow 33.6\%$).
>     * This confirms that while the modulator provides frequency content, only BioRNN’s stabilized wave dynamics can effectively "resonate" with and utilize these patterns.
>
> **2. Re: Evidence for "Long-standing Instability" and Necessity of Mixed Discretization**
>
> The reviewer asks for citations regarding the difficulty of training physical wave equations. We appreciate this opportunity to clarify the precise scope of this claim and the necessity of our approach.
>
> * **Clarification of "Instability":** We acknowledge that the "long-standing instability" refers fundamentally to the **Courant-Friedrichs-Lewy (CFL) condition** [1], a well-established and unforgiving constraint in computational physics, rather than a specific debate within the RNN literature.
> * **Novelty & Training Difficulties:** To our knowledge, there are no prior works explicitly documenting "failed gradient-based training of FDTD-RNNs" **precisely because we are the first to attempt embedding a direct, explicit 2D Finite-Difference Time-Domain (FDTD) solver as a trainable recurrent transition function.** Prior bio-inspired models (e.g., NWM, wRNN) circumvent this instability by using closed-form analytical solutions or coupled oscillators, avoiding the FDTD formulation entirely.
> * **Experimental Evidence:** Our internal experiments confirmed why this gap exists: standard FDTD schemes (Forward/Central) lead to **immediate gradient explosion (NaNs)** when the wave speed $c$ and damping $k$ are learned via backpropagation. This occurs because the optimizer inevitably pushes parameters into the unstable region, violating the CFL condition.
> * **Contribution:** BioRNN fills this gap. We introduce the **Mixed Finite-Difference Scheme** not just as an improvement, but as the necessary **stabilizing mechanism** (via the numerical dissipation proved in Lemma 1) that makes this unexplored class of FDTD-based RNNs trainable for the first time.
>
> **3. Re: Biological Motivation (Wave Equations & Neuromodulators)**
>
> We appreciate the opportunity to clarify the biological links. We will include the following citations in the revision:
>
> * **Why Wave Equations?** The brain is not merely a set of isolated coupled oscillators; neural activity propagates spatially across cortical sheets as **traveling waves**. The 2D wave equation serves as the macroscopic continuous limit of these local synaptic interactions.
>     * *Citation:* **Muller et al. (2018)**, "Cortical traveling waves: mechanisms and computational principles" (Nature Reviews Neuroscience), highlights that traveling waves are a fundamental computational mechanism for spatial gating and information flow.
>     * *Citation:* **Sato (2022)**, "Cortical traveling waves reflect state-dependent hierarchical sequencing," links these wave dynamics to sequence processing.
> * **Link to Neurotransmitters:** In biological circuits, neuromodulators (like acetylcholine or dopamine) often do not carry sensory content themselves but alter the **excitability and gain** of neural populations.
>     * Our **Input Modulator** is **conceptually inspired** by this mechanism. It abstracts the biological principle of "gain modulation" into a computational operation: by transforming the input spectrum to trigger specific resonant modes in the grid, it effectively shifts the network's processing state without altering the underlying topology. This serves as a functional analogy to how neuromodulators guide cortical circuits.

---

> > ### Author Response · Authors · 2025-11-18
> >
> > **4. Re: Ablation of Input Modulation**
> >
> > The reviewer asked for an ablation of the modulator specifically.
> > * **Response:** **Table 2** serves as this direct ablation study.
> >     * **Left Column (Base BioRNN):** Shows performance *without* modulation (e.g., 11.2% on sMNIST).
> >     * **Right Column (BioRNN):** Shows performance *with* modulation (e.g., 98.1% on sMNIST).
> >     * This comparison isolates the modulator's contribution, showing it is critical for tasks where the raw input lacks intrinsic oscillatory structure (like static pixels in sMNIST).
> >
> > We hope these clarifications regarding the existing comparisons in Table 2 and the foundational stability challenges address the reviewer's concerns.
> >
> > *[1] Courant, R., Friedrichs, K., & Lewy, H. (1928). On the partial difference equations of mathematical physics.*

---

### Meta-Review · Area_Chair_bdHM · 2025-12-30

**Summary:**

The paper proposes a *Biologically Inspired RNN* (BioRNN) that is based on two steps: a) an oscillatory modulation of the input and b) RNN hidden layer is instantiated in terms of discretized solutions of a class of wave equations. The authors stably discretize the wave equation in terms of a suitable spatio-temporal discretization that allows for stable training. They present experiments with a sMNIST, noisy CIFAR and ESC-50 datasets. The idea of the paper is interesting and has been acknowledged as such by the reviewers. However, the main weaknesses of the paper, based on an independent reading by the AC and also repeatedly emphasize by the reviewers, are

[1.] Very thin neurobiology link -- it is totally unclear why a wave equation on the form proposed by the authors is used for neural activity in the brain. On being asked by one of the reviewers, the authors provided tangential citations about *cortical traveling waves* without in any way being able to link their specific form of the linear wave equation as the source of these traveling waves. The authors should be aware of the fact traveling waves can arise in a variety of PDEs starting from reaction-diffusion equations (mostly likely models of neural activity) to nonlinear hyperbolic equations, Thus, *biological plausibility* of the paper is very thin, if it exists at all.

[2.] The empirical performance of the model is poor. The authors considered a very limited set of experiments on relatively easy datasets and their model is not even performing well on any of them. It is worse than plain vanilla GRU on every experiment, raising the question on why the model is of any practical utility to begin with.

[3.] The discretization of the wave equation that the authors propose is pretty standard numerical analysis although it is claimed as a major
contribution.

[4.] The authors do not provide a careful analysis of the training and inference costs of the model, vis a vis baselines.

Given these weaknesses, the paper cannot be accepted in its current form.

**Reviewer Concerns:**

The authors have addressed some of the technical questions of the reviewers but the main concerns that have not been satisfactorily addressed are

[1.] Biological Plausibility

[2.] Empirical Performance.

**Reviewer Scores:**

Reviewer QrBi has scored the paper as a reject. Their main concerns on biological plausibility and empirical performance have not been addressed satisfactorily and it is unlikely that the reviewer would have raised their score.

Reviewer U5uu's main concern was also about weak empirical performance of the model. Given the authors' rebuttal, it is unlikely that they would have raised their score of Reject.

Reviewer  LSat had scored this paper as borderline Accept but with a very low confidence. Their concerns were minor but a clear question about the training and inference time that was not clearly answered.

Reviewer do4L provided a borderline Reject with moderate confidence. The reviewer pointed out some technical/positioning weaknesses that were addressed by the authors, making a slight increase in the score likely.

But the overall recommendation of the reviewers backed the independent judgement of the AC in rejecting the paper based on the grounds outlined above.

---

### Decision · Program_Chairs · 2026-01-26

Reject